# Barriers and Levers for the Implantation of Sustainable Nature-Based Solutions in Cities: Insights from France

**Chloé Duffaut** [1,†], **Nathalie Frascaria-Lacoste** [2] **and Pierre-Antoine Versini** [1,*]

1 Hydrology Meteorology and Complexity Laboratory, École des Ponts ParisTech, 77455 Champs-sur-Marne, France
2 Université Paris-Saclay, CNRS, AgroParisTech, Ecologie Systématique Evolution, 91190 Gif-sur-Yvette, France
* Correspondence: pierre-antoine.versini@enpc.fr
† Chloé Duffaut Left the Hydrology Meteorology and Complexity Laboratory, École des Ponts ParisTech, on 1 September 2021.

**Abstract:** The challenges of the 21st century, namely, climate change and loss of biodiversity, especially present in heavily populated areas, should be addressed. Nature-based Solutions (NBS) seem to offer a suitable answer to these challenges. However, this new concept is not always easy to implement in a sustainable manner. In an effort to identify the barriers and levers for the implementation in cities of these sustainable NBS, several professionals working on them in France were interviewed. The first analysis with the most quoted words shows that the constraints would be mainly related to technique, management, biodiversity and people. The levers would be related to projects, roofs, people, legislation and services. Further analysis shows that most of the interviewees feel that the main barriers are the lack of knowledge, political will, financial resources and regulations. There are also cultural limitations, such as the use of exotic horticultural plants rather than wild local species. According to them, the technical problems should be easy to solve. To address these issues, the interviewees propose the development of transdisciplinary research disciplines, as well as on-field collaboration between all NBS actors in cities. They also recommend specific funds for NBS and their implication in related regulations. Demonstrative examples of urban NBS highlighting their multiple benefits are also needed to encourage their replication or upscaling. Education and communication are essential to shift the traditional points of view on nature in the city.

**Keywords:** nature-based solution; city; barriers; levers; interview

## 1. Introduction

Today, urban environments are facing many challenges, including global warming, loss of biodiversity and all their consequences. Within this context, Nature-Based Solutions (NBS) appear to be the ideal tool. Indeed, they can act on both fronts. The International Union for Conservation of Nature (IUCN) defines NBS as follows: "actions to protect, sustainably manage and restore natural or modified ecosystems that address societal challenges effectively and adaptively, simultaneously providing human well-being and biodiversity benefits" [1]. The European Commission defines NBS as the following: "Solutions that are inspired and supported by nature, which are cost-effective, simultaneously provide environmental, social and economic benefits and help build resilience. Such solutions bring more, and more diverse, nature and natural features and processes into cities, landscapes and seascapes, through locally adapted, resource-efficient and systemic interventions." [2]. These two definitions are often used as references, but others also exist. Indeed, there are many definitions of NBS, but most agree that these solutions are beneficial to the environment and to humans rather than just focusing on the restoration of nature and its conservation [3]. The following examples can be listed: restoration of mountainous slope forests, which help avoid erosion and landslides; preservation of mangroves to limit the risk of submersion; revegetation of buildings to improve their thermal properties.

The NBS concept is mainly European and is often associated with climate change, urbanization, water management, urban heat islands, air pollution, well-being, human health and sustainability [4]. The concept is recent and is still emerging [5,6], with many related scientific publications. In a search performed by the authors on 10 June 2021, Scopus found 975 documents with titles, abstracts and keywords containing the term "nature-based solutions". Under the topic of the urban environment, the usual cited NBS are as follows: urban forests [7,8], green roofs and walls [9,10], ecological corridors [11], or green swales [9,12]. They are usually studied to manage stormwater issues or to mitigate urban heat islands.

It is worth noting that some scientists, especially in the field of nature conservation, treat NBS as "yet another buzzword" [13]. Even if the concept is currently a hot topic, it is challenged by its physical implementation, which is often complicated by many factors [14]. These difficulties are particularly noticeable in the urban environment due to its typical dense nature. Indeed, the implementation of NBS in urban settings raises the following number of operational questions: How to adequately integrate them into the concerned context? How to evaluate their costs and benefits? How to design solutions to meet the different challenges? etc. [15]. In addition to these issues, there are many other barriers facing the implementation of sustainable NBS in cities. With the help of a dozen people interviewed working on NBS in France, this study aims to identify these barriers and highlight the levers that can be used to overcome them. By responding to this problem, answers for subsequently upscaling NBS in cities can be obtained.

## 2. Materials and Methods

For this study, several actors in France working on NBS were interviewed in a semi-structured way. The interviewees were selected because the authors knew that they or their organizations/companies were working on urban NBS in France for many years. Hence, they were able to share relevant and eclectic discourses, experiences and analyses. The authors are aware that this selection may have led to a bias, but as this work is rather qualitative, this bias may be negligible. This study allows for the identification of barriers and levers by "experts" in the field and is not a quantitative study on the level of knowledge of the general public. In order to cover the whole spectrum of specialties, interviewees from academic, institutional and operational circles were chosen, ensuring the same number of people in each category. The profiles of these interviewees are listed herewith the following: Ecologist and project leader at the Regional Biodiversity Agency, researchers in Hydrology or Ecology at the Centre for Studies and Expertise on Risks, the Environment, Mobility and Urban Planning (CEREMA), the French Research Institute for Development (IRD) and the Museum of Natural History, general manager and head of the planning and natural environment department of the Intercommunal Union for Hydraulic Development of the Croult and Petit Rosne Valleys, project manager at the IUCN, independent agro-economist and project manager at the Gally design office, project manager in a design study at Topager, technical manager at SOPRANATURE (green roofs and facades of SOPREMA, building, waterproofing and insulation company), and a manager of the EcoQuartier mission (EcoQuartier is an approach supported by the French Ministry of Ecological Transition, which promote new ways of designing, building and managing the city sustainably. It includes 20 commitments, a label, a club to meet and tools to train). More details are provided in Table 1.

**Table 1.** List of interviewees with their reference number, function, organization, French acronym and website of their organization, sector of activity and the date they were interviewed. In the rest of the article, the letters in brackets correspond to the reference letters in this table to indicate, which interviewee is cited.

| Letter | Function | Organization | Acronym | Website | Sector | Date |
|:---:|:---:|:---:|:---:|:---:|:---:|:---:|
| A | Head of studies | Regional Biodiversity Agency | ARB | https://www.arb-idf.fr/ (accessed on 20 July 2022) | Institutional | 17 November 2020 |
| B | Regional animator | Regional Biodiversity Agency | ARB | https://www.arb-idf.fr/ (accessed on 20 July 2022) | Institutional | 27 November 2020 |
| C | Researcher in hydrology | Center for studies and expertise on risks, environment, mobility and development | CEREMA | https://www.cerema.fr/fr (accessed on 20 July 2022) | Academic | 4 December 2020 |
| D | Head of the urban planning and natural environment department | Mixed syndicate for the hydraulic development of valleys | SIAH | https://www.siah-croult.org/ (accessed on 20 July 2022) | Operational | 8 December 2020 |
| E | NBS project manager | International Union for Conservation of Nature | UICN | https://uicn.fr/ (accessed on 20 July 2022) | Institutional | 14 December 2020 |
| F | General manager | Mixed syndicate for the hydraulic development of valleys | SIAH | https://www.siah-croult.org/ (accessed on 20 July 2022) | Operational | 16 December 2020 |
| G | Research director in ecology | Institute of Research for Development | IRD | https://www.ird.fr/node/8 (accessed on 20 July 2022) | Academic | 7 January 2021 |
| H | Project manager | Gally's design office (design, plants, urban biodiversity and agriculture) | GALLY | https://www.lesjardinsdegally.com/agence/le-bureau-detudes-de-gally (accessed on 20 July 2022) | Operational | 20 January 2021 |
| I | Project manager/Head of mission | Topager (Edible and wild urban landscape)/Museum of Natural History | TOPAGER/MNHN | http://topager.com/ (accessed on 20 July 2022) https://www.mnhn.fr/fr (accessed on 20 July 2022) | Operational/Academic | 3 February 2021 |
| J | Researcher in ecology | Center for studies and expertise on risks, environment, mobility and development | CEREMA | https://www.cerema.fr/fr (accessed on 20 July 2022) | Academic | 4 February 2021 |
| K | Technical manager | SOPRANATURE (Vegetation system) | SOPREMA | https://www.soprema.fr/fr/nos-produits/vegetalisation/sopranature (accessed on 20 July 2022) | Operational | 19 February 2021 |
| L | Responsible of EcoQuartier | Ministry of Ecological Transition | MTE | http://www.ecoquartiers.logement.gouv.fr/ (accessed on 20 July 2022) | Institutional | 4 March 2021 |

A questionnaire was especially prepared for this purpose. Most of the questions were developed by the authors; however, the first Author adapted some of the questions to fit the profile of each interviewee. The questionnaire was divided into the following six parts: (i) context of the interview, (ii) personal information, (iii) NBS in the urban environment, (iv) biodiversity and the ecosystem functions of these solutions, (v) constraints, barriers and levers for the implementation and the sustainability of these solutions, and (vi) perspectives. The main questions about barriers and levers are the following: "In your opinion, what could be the obstacles to the implementation and sustainability of NBS in cities?" and "What are the levers that could be used to promote the establishment and sustainability of NBS in cities?". For more details on the questions, see the "Questionnaire used during interviews" in the Supplementary Material.

The semi-structured interviews were conducted in French by the first author via videoconference techniques using the Meet Jitsi website (https://meet.jit.si/; accessed on 20 July 2022) at the following address: https://meet.jit.si/EntretiensSolutionsFondeesNature (accessed on 20 July 2022). Each interview lasted between 45 and 90 min. They were carried out between 17 November 2020 and 4 March 2021. The first hour of the interviews was recorded (image and sound) directly via the Meet Jitsi site and saved on the Dropbox cloud. The complete interviews were recorded (sound only) via a voice recorder on a digital tablet. In addition to handwritten notes taken during the interviews, full transcripts were made from the video and audio recordings.

First, word clouds were created with the answers translated into English to the questions on constraints and barriers on the one hand and on levers on the other hand. This enabled us to highlight the words most used by the interviewees in their answers (the more often a word is quoted the bigger it appears in the cloud). The word clouds were made with R and Iramuteq software using the active forms of the words (verbs, nouns, adjectives) cited at least 5 times for barriers and 3 times for levers. Words that did not provide information were removed (put, lot, thing, good, case, addition, false, feel, today, small, real, relate, back, subject, start, true, set, show, fact, begin, end, sense, remain, bite, area, bring, percentage, type, part, talk, necessarily). Words of the same family were merged.

Then, a deep analysis of the results of the interviews was conducted to explain in detail the main identified constraints and levers. In order to illustrate the different arguments, put forward by the interviewees, verbatim their answers are inserted in the presentation of the results, and the related discussion. Illustrating a personal opinion, these verbatims shed light on a specific topic and provide relevant information that can be confronted with bibliographic references. The results of the interviews are divided into two parts. The first part lists the constraints and difficulties encountered by the actors in implementing Nature-Based Solutions. The second part gives the levers for effective implementation under a framework of sustainability.

## 3. Results and Discussion about Barriers to the Implementation of Sustainable NBS

According to the word cloud (Figure 1a), the constraints to urban NBS are various. The most highlighted words can be related to categories discussed in the following Sub-Sections: technique (Section 3.1.2), management (Section 3.1.3), biodiversity (Section 3.3.2), people (Section 3.3.3), etc.

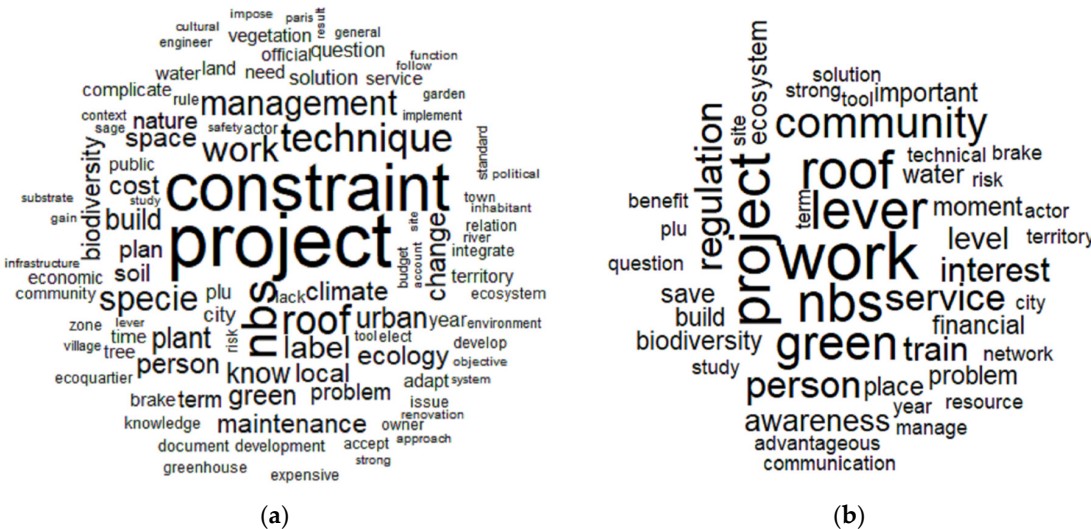

**Figure 1.** (**a**) Word clouds with answers to the questions on constraints and barriers to the implementation and sustainability of nature-based solutions (NBS) in cities; (**b**) word clouds with answers to the questions on levers to the implementation and sustainability of NBS in cities. The larger a word appears, the more it has been quoted.

### 3.1. Knowledge and Technical Barriers

3.1.1. Confusion Due of Multiplicity of Terms and Lack of Knowledge

The interviewed ecologist points out a problem, which is the *"multitude of concepts that designate more or less the same thing, that come together to designate nature-related elements: nature-based solutions, green and blue solutions, green and blue webs, nature in the city, ecological engineering, urban restoration, green infrastructure, etc."* (A). The same person adds the following: *"I think what you call NBS, is not called so by the people who implement them, especially in urban areas. NBS is a new concept, not very common in communities, they'll talk more about a landscape project, a wetland or a retention pond, a park or a garden, etc."* (A). For him, this term is mainly used by scientists. The problem of NBS-connected terms, even synonymous ones, has already been raised in the scientific literature [16]. The multiplicity of terms can be counterproductive, as it makes research more difficult to conduct. Accordingly, there is a risk of misunderstanding due to contested nomenclature that might lead to a replication of efforts. Moreover, the IUCN project manager said that the term NBS *"[is] not always well-used (one of my roles is to re-define it to ensure that it is used well)"* (E). It is therefore necessary to recall the adopted definition to homogenize the discourse among different stakeholders.

Several interviewees pointed out that there is still a lack of data and knowledge on NBS in urban settings. Even if studies on the subject have multiplied in recent years, there are still grey zones that need to be filled [17,18]. Indeed, ecosystems and the urban environment are complex, and their interrelationship is difficult to analyze. Thus, soils are often underrepresented and tools enhancing soil quality are needed, according to one interviewee. An interviewed researcher in ecology said that it was necessary to *"increase scientific knowledge on the components of biodiversity that allow to maximize ecosystem functions and services"* (J). According to the literature, there is little evidence of the multiple benefits of NBS [3,19].

A project leader on NBS said the following: *"There is a lack of indicators that reveal the efficiency of NBS. [ . . . ] It is difficult to evaluate the gains resulting from NBS, especially when taking into account all the costs and benefits, and not just the targeted problem. It is accordingly hard to mobilize stakeholders, due to the lack of concrete proof on returns."* (B). According to Kabisch et al. [18], there is a need to develop indicators that incorporate environmental performance, health and well-being, citizen engagement, and maintenance/transferability. People who implement NBS need to be reassured that these solutions will deliver the

expected benefits; otherwise, investing in them might be useless. It is also important to mention that long-term studies (i.e., over large time spans) on NBS in cities are still too few, especially those dealing with the longevity of the advantages brought by these solutions.

### 3.1.2. Technical Problems That Can Often Be Overcome

Despite a general lack of knowledge, many interviewees reported that technical issues were not the most difficult elements to solve. In fact, most of the time, technicians know how to eliminate these types of constraints thanks to their own knowledge and to their field expertise, as illustrated by an ecologist as follows: *"Yes, there are technical constraints, but they can be easily overcome, if you do things in the right order with specialists it can be done."* (A). There is even a technical handbook specific to NBS that can help people work on them [20]. However, not all technical constraints are easily overcome. While this might be the case for technical constraints tied up to buildings, the technical constraints linked to ecology are harder to solve. Indeed, as observed, there is a definite lack of knowledge and skills in this particular field. As a guideline, a hydrologist told the authors that simplicity should be sought as follows: *"Simpler solutions should be developed. Currently, NBS comprise of a lot of things: some things are very technical and others are more elementary. In my views, the more elementary solutions are closer to NBS and are better than the very technical aspects."* (C).

### 3.1.3. Lack of Maintenance and Durability

The ecologist of the Regional Biodiversity Agency said, *"There is usually a lack of monitoring over time, that is to say a before/after assessment. Also, there is a lack of understanding on whether a gain for biodiversity has been attain and whether certain ecological functions were affected, [ . . . ] to make an annual follow-up for instance is difficult, [ . . . ] Similarly, the fear from part of communities exists based on not knowing what works over a long term."* (A). The hydraulic union manager for the natural environment (D) gave the example of a watercourse that was reopened in 2014, where only a single fauna and flora inventory had been carried out. According to the manager, it would have been better to carry out a survey every two years at least. She also explained that long-term monitoring is important to determine, for instance, *"the behavior of the stream after dry weather, or rainy weather, and see how the vegetation manages to recover."* (D). She added that it was tricky to find *"a good boundary between the desire for a natural stream and the reality of urban area."* (D). NBSs are partly made up of living beings and therefore are in constant evolution, thus requiring continuous maintenance. However, this maintenance is not always taken into consideration in NBS projects [18]. The problem of lack of monitoring and/or maintenance may be related to the fact that administrations have a short-term view when a long-term one should be favored for NBS [21,22]. The problem of maintenance raises the following number of questions: *"Who will do the maintenance? How do we make sure we have sustainable funding for this maintenance?"* (E).

Although maintenance can be seen as a barrier to NBS, it can also represent an opportunity, as one person from the water development union (D) explained. Indeed, the projects on which they work must manage rainwater at the plot parcel level. Currently, in the case of underground water basins, maintenance is performed only in the first few years. Then, because the basins are not visible, maintenance tends to drop. So, the union favors the management of open-air rainwater, for which it is easier to implement a perennial maintenance program and is prone to the creation of NBS such as wetlands.

### *3.2. Contextual Barriers*

### 3.2.1. The Challenge of Adapting NBS to Local Climate and Climate Change

In the questionnaire, the climatic factors are considered constraints because they require a significant adaptation of NBS (e.g., choice of plant species). Such a task is not always easy and requires considerable knowledge. Ideally, it is better to use native plants adapted to the local climate and to adapt the plant range on a case-by-case basis. However, for practical reasons, this is not always possible.

In addition, climate change and its consequences must be taken into account, and species that can cope with them must be found in the near future [16,23]. However, this is not an obvious task, as the project manager of the Gally design office said the following: *"If I plant trees, they must still be there in 20 or 30 years. Hence, how do I select them?"* (H). A researcher asked the following similar question: *"In anticipation of a change in climate, shouldn't we start considering species that are not necessarily local, but more adapted to periods of drought (e.g., Mediterranean species)?"* (J). Even if certain people try to find solutions to answer this problem, it is not yet the norm and many more efforts are still needed.

To limit greenhouse gases and therefore mitigate climate change, it might be interesting to consider short-term actions, for example, trying to use local materials for the substrate of green roofs. In this regard, a seller of green roofs explained that this is something that can be very complicated to do in practice. Indeed, it is necessary to *"[extract] the soil at the right time during the construction phase, [characterize it] in the laboratory, [use it] partly on a roof [after having it mounted]."* (K). Mounting the soil is just as complicated, as follows: *"You have to put it in big bags, which are very expensive. This means that you have to set up a bagging workshop on site or you have to move the local material onto a platform with semi-trailer trucks to use this material, bag it as is or slightly rework it with lightening or structural elements, to bring it back to the site for use."* (K). This leads to having to rethink the implantation/construction methods.

### 3.2.2. Too Little Space in the City and Land Prices Are Often High

Something that has not been mentioned much, but which is a characteristic reality of the urban environment, is the lack of space and the price of land which can be very costly in some cities such as Paris. Indeed, the average built density ((footprint × building height)/study area) in Paris is equal to 2; in comparison, the built density of an individual housing operation is about 0.3 [24]. In April 2021, the average price of a square meter in Paris was EUR 10,780 [25]. This problem does not concern green roofs as they are not in direct competition with other infrastructures because they can be inserted on the top of buildings. However, the land characteristic is a major problem for ground NBS. The city is a built environment that can be very dense and NBS can require large spaces that are difficult to find in such an environment [26]. This lack of space can even lead to the redesign of an urban project (D). On this subject, an interviewee points out that NBS can take up a little more space than traditional solutions. He added the following: *"Today, we are already struggling to get people to accept more open space in development projects, more green spaces or land. Projects are often far too dense. So, there's an issue of creating space for these NBS."* (A). The price of land can also be a disadvantage for the implementation of NBS and sometimes conventional infrastructures can be more profitable, especially in the short term. This competition for land use has already been discussed in the literature [18].

### 3.3. Cultural Barriers

### 3.3.1. Greenwashing: NBS Are Not as Eco-Friendly as Portrayed and Failed Cases of NBS Give the Impression That NBS Are Not Efficient Solutions

One of the risks of environmental projects is greenwashing. Indeed, sometimes the ecological aspect of certain projects can be oversold and overused by the NBS concept term [27]. The project leader at the regional biodiversity agency thus refers to *"NBS can be used indiscriminately, for example referring to false ecological engineering"* (B). One researcher explains it the following way: *"The risk with green roofs is that they oversell ecological interests [ . . . ] services and benefits that might never be provided at the end."* (G). In this context, green roofs can be considered mono-specific lawns that represent little interest in biodiversity. Thus, many private NBS commitments are presented as offsets, which often imply greenwashing [28].

Failed examples are also a major issue. The general manager of the water management union said that there were bad examples in which NBS *"brought disadvantages and increased risks"* (F). He made particular reference to bad practices in a reopened river zone, such as *"uncontrolled picnics spots, quads, and motocross activities"* (F) even the accumulation of waste,

and the excessive consumption of alcohol. When NBS planners observe such effects, they might reconsider the implementation of their projects.

### 3.3.2. Biodiversity Is Not in the Foreground of Our Societies

Generally, biodiversity and its erosion are not in the foreground of our societies. For example, in the United States, most people prioritize other issues such as terrorism, health or the economy [29]. Presumably, these concerns are the same in Western societies such as France. The recent health crisis due to COVID-19 has undoubtedly increased health concerns and consequently pushed the biodiversity crisis into the background [30]. The economy is also often at the center, leaving little space for environmental issues [31]. The person responsible for urban planning and the natural environment of the hydraulic union confirms that in urban development, biodiversity is not the priority, as follows: *"Designing a housing project with a focus on biodiversity as a priority can be complicated, since it comes second."* (D). Even within environmental issues, the project manager of the Regional Biodiversity Agency told the authors that protecting biodiversity is not a priority as follows: *"[the Ministry is really focused] on adapting to climate change, climate change in general and renewable energies."* (B). Biodiversity often takes a back seat behind aesthetics (or agronomy), as one ecologist noted the following regarding NBS substrates: *"Today, substrates are made by landscapers who want to give an agronomic or ornamental aspect to their project. In any case, it does not consider the type of plants* i.e., *wild plants, or local plants, etc. [ . . . ] And that's an issue in NBS: to move towards more local plants."* (A). Another ecologist noted, *"The fact that there are neither landscapers, nor ecologists on the project, can raise the question of how someone (i.e., the architect) who does not necessarily have this eco-friendly culture is going to put in place everything."* (I). Indeed, the implementation of NBS in cities is often performed through an opportunistic approach (A).

According to someone from the hydraulic union, despite what was presented, biodiversity seems to be increasingly taken into consideration, particularly in the legislations of urban planning. However, as she explained, this approach is often more oriented towards humans as follows: *"Concerns today about the impact of projects, the long-term vision of a site, the integration of nature, having green corridors, having less concrete, and more vegetation remains within the framework of a vision of an environment adapted to humans: "You are going to be in a neighborhood with lots of trees"."* (D).

### 3.3.3. Social and Cultural Barriers Are Often Predominant

An ecologist declared the following: *"I would say the first constraint to implementing NBS is cultural. It's the fear of using nature versus grey infrastructure. [ . . . ] We're afraid of these solutions because we don't find them reliable."* (A). The hydrologist (C) also mentioned the fear that NBS may not work, which has been documented in literature such as the fear of lower performance of green infrastructure versus grey infrastructure [32]. This cultural barrier seems to be present at all levels, whether from the general public, communities, urban planners, etc. Thus, there is a lack of citizen awareness, support and interest in NBS [22,33].

Later, the ecologist added the following: *"People are afraid of nature. [ . . . ] People still have a rather negative relationship with nature, even if it is changing. We see it in the case of the wetland in Vignois, there are complaints from inhabitants about mosquitoes. It's a daily job to try and get people to accept it."* (A). This fear of nature may relate to the fear of the unknown discussed by Kabisch et al. [18] in the face of uncertainties and risks of implementing NBS in cities, as well as the changes these may induce in urban planning. This fear of nature can also relate to real problems called ecosystem disservices, such as the mosquito bites mentioned hereabove. Indeed, ecosystem disservices are inconveniences caused by nature and they can be diverse in cities [34].

The manager of the urban planning and natural environment in the hydraulic union mentions many human-related barriers as follows: *"Often statements such as "Biodiversity is very good, but not in my place!" are common. From the moment when vegetation is allowed to grow, having a height of 50 cm of vegetation [ . . . ] in an urban environment is not something acceptable,*

*because it does not look clean. [ . . . ] We try to make something beautiful so that it becomes better accepted. [ . . . ] We can also have complaints because of pollen and its associated allergies"* (D).

Another aspect of social barriers is the cost of organizing services or changing roles within communities (C).

### 3.4. Institutional Barriers

#### 3.4.1. Too Little Funding for NBS

There are also economic constraints, including the uncertainty about the cost of NBS. For example, one hydrologist (C) said that NBSs are perceived as more costly than conventional solutions. Such observations are also found in the literature on green infrastructures [32]. The technical manager of SOPRANATURE told us, *"The economic constraints of the project mean that green spaces are the fifth wheel. When there are savings to be made, they are found in green spaces. The same applies for building, when savings are to be made, it is often vegetation roofing that will suffer."* (K). The ecologist from the Biodiversity Agency told us the following on this topic: *"We are afraid of having to manage ecosystems for too long. Whereas with good grey infrastructure, we know the cost, at least for the short term."* (A). His colleague added the following: *"Big problem to obtain financing complicates the mobilization of the public on NBS, in particular because NBS are very broad, [ . . . ]. Many terms already used before. So, funders are wondering what NBS have to add."* (B). She also says it is hard to obtain a quick return on investment. According to the European Commission's 2015 report [35], the cities' budgets for green spaces are very small. Specifically, the lack of dedicated funding for NBS implementation in cities has already been highlighted [36], and financial incentives to use NBS are also missing [37]. According to Toxopeus and Polzin [38], the main financial barriers to urban NBSs are lack of coordination between public and private funding and a lack of integration of NBS benefits into valuation and accounting methods.

#### 3.4.2. Too Little Space for NBS in the Regulations

Regulations can be a constraint for the development of NBS in the city because, as one interviewee points out, *"planning documents today do not have a space for these NBS."* (A). He cites the PLU (i.e., the Local Urbanism Plan) and the SCoT (i.e., the Territorial Coherence Scheme). An NBS officer (B) gave an example that an NBS was to be implemented in a zone labeled as "to be urbanized" in the PLU, and that the project in which she was involved could fund NBS only in zones classified in the document as natural. This example shows how regulations can become obstacles to the implementation of NBS in cities. For now, French regulations do not sufficiently encourage these practices. Kabisch et al. noted back in 2016 [18] that urban administrations may lack information on legal instruments and requirements for implementing NBS.

#### 3.4.3. Lack of Political Will

Another important obstacle facing NBS in cities is the lack of political will. Indeed, the development of such solutions requires political initiatives from elected officials, but that is not always the case. This lack of political will as a major constraint to the implementation of NBS in cities was also identified by Sarabi et al. [22]. Moreover, the representative of the Biodiversity Agency (B) told that even if the political will was to be there, municipalities and elected officials can change quickly and be replaced by some people less concerned by NBS. This can compromise projects undertaken during previous mandates. In France, municipal councils and mayors have a six-year mandate [39]. In comparison, the implementation of an urban project takes at least ten years, often much longer than a political mandate [40]. The problem of changing administration was also mentioned by Kabisch et al. [18]. Davies et al. [35] also discussed the long-term vision for green spaces that must be modified due to policy changes. *"It would be necessary for municipalities or agglomerations to have a real desire, to be the driving force, in the PLU to reinforce the presence of biodiversity component on their territories"* (D), said a person from the hydraulic union.

**4. Results and Discussion about Levers to the Implementation of Sustainable NBS**

According to the word cloud (Figure 1b), the levers for urban NBS are numerous. The most highlighted words can be related directly to the categories discussed in the following next sub-sections: projects and roofs (Section 4.1.3), communities and people (Section 4.2.2), regulations (Sections 4.3.3 and 4.3.4), services (Section 4.2.1), etc.

*4.1. Raising Awareness through Knowledge*

4.1.1. To Address the Lack of Knowledge, Research and Diagnostic Efforts

To respond to the lack of knowledge of NBS, more research is required. The risks must be diagnosed in detail in order to identify the sectors that are in most need of NBS. It could also be interesting to develop *"territorial diagnostic tools, for example, to know where the risks of flooding or heat waves lay, and/or to target the different risk levels."* (A). An ecologist cites other avenues of research as follows: *"defining the NBS scale, or what is grouped behind it, and the position of biodiversity within it."* (A). To maximize the contribution of biodiversity, more knowledge of species ecology is required in order to create for them favorable conditions. It would also be interesting to evaluate the real monetary costs and benefits of an NBS implementation in the city. According to Kabisch et al. [18], the areas of knowledge to be developed are the effectiveness of NBS, the relationship between NBS and society, the NBS design, and its implementation. There are already several research projects on urban NBS, such as the European Horizon 2020 "REGREEN" project. The main research axes of this project are the following: improving knowledge of NBS, the development of mapping and modeling tools and the study of the links between well-being, health and nature in cities [41]. There is also the GROOVES study (Green ROOfs Verified Ecosystem Services) carried out by the Regional Biodiversity Agency, which applies to green roofs in the Paris region. In this study, inventories of flora and fauna were carried out and ecosystem services such as water retention were studied. In general, more weight should be given to NBS studies [42]. To address the lack of follow-up indicators on the NBS, IUCN has developed a Global Standard with 3–5 indicators per criterion in the form of a traffic light (E).

Beyond research, the operational people who set up NBS in cities should make a diagnostic effort, taking into account the context as a whole (local climate, context and coherence). For example, the flora in the vicinity where NBS is to be set up should be studied, and possibly plant the same into the NBS. One of the ecologists (A) associates this with ecological engineering, which should be a reflex when setting up NBS.

4.1.2. Formation and Education on Nature and NBS at All Levels

To go against the cultural barriers, the director of the hydraulic union explains as follows: *"We are in a logic of training and not of communication; this is not enough to fight prejudices, caused by for instance excessive services of green spaces (over shaving of banks under the responsibility of the union). Afterwards, we must not be surprised to witness erosion on the banks"* (F). One of the ecologists (A) interviewed insisted that elected officials, technical services, schools, communities and private sector individuals must be trained on NBS issues and ecology, in general, to become well informed on these topics and therefore to implement successfully NBS. In this dynamic, the director of the hydraulic union talked about the following program, initiated by them: *"an educational program that we introduced this year into riparian schools"* (F). The ecologist (A) also believes that engineering firms need more technical training. Indeed, he noted that in most cases ecological compensation measures were badly implemented by engineering companies as they lacked the necessary skills. According to this ecologist, landscapers and architects must be trained in NBS too, so that they can also implement them correctly. Indeed, it would be good if professionals working on urban infrastructure were trained in NBS and not just in grey infrastructure [42]. On this subject, a researcher in ecology adds the following: *"[We must] rethink the training of public works which are still under the influence of large groups historical lobbying for such as the "all pipe" for stormwater management lobby, "all to the sewer" lobby or any classic schemes that manage the networks."* (J). None of the interviewees specified whether these education

programs should be developed at the pre-service or in-service level, but presumably, they should exist at both levels. Education programs on NBS and related topics at many levels have already been mentioned as a lever for their implementation [3,43]. Thus, through education, people can be motivated to protect the environment [44]. This education can take many forms; for example, in Poland (in Katowice), a local community that trains youth in sustainability was the main actor in organizing and networking citizens and the city for a food festival where the term "nature-based" solutions was explained [10].

4.1.3. Demonstrator Examples That Can Be Replicated

It is important to have good examples to replicate. Indeed, the following is one of the seven lessons about urban NBS that Frantzeskaki [10] drew from her study: the need to learn about NBS and replicate them over the long term. An ecologist stated the following: *"I think we need large demonstrators today, such as large wetlands, or experiments like in Lyon, on rue Garibaldi, on the cooling effect of trees, which show us concretely what benefits these NBS bring, and to be able to quantify them."* (A). According to another interviewee, it is necessary *"to bring the elected representatives on the site"* or *"project owners should come and see what is being done"* (F). One NBS project manager summarized it the following way: *"You need to have examples, [which is] needed by communities. Knowing that another community had the same problem, seeing what has been done on their site and the positive effects of NBS application and seeing that it works, makes you want to do the same. [We need to] get the momentum going."* (B) A hydrologist said the following: *"[We must] manage to show that NBS work, that they are good at absorbing rain, that in places with nature, people are happier and to show that NBS are sustainable"* (C). People likely to uptake NBS need to be reassured. If they see NBS functioning properly and providing ecosystem services, they may wish to replicate them. For the Life ARTISAN project (Increasing the Resilience of Territories to Climate Change by Encouraging Nature-Based Adaptation Solutions, https://www.ofb.gouv.fr/le-projet-life-integre-artisan; accessed on 20 July 2022, there are 10 demonstration sites, including two in the Ile-de-France region (Les Mureaux and AQUI'Brie), which have been selected on the basis of the NBS project, to demonstrate the implementation of nature-based adaptation solutions. Demonstration sites provide an opportunity to evaluate NBS in practice and adapt their management approach [18]. The European Union has invested heavily in such NBS demonstration projects, notably with the Horizon 2020 research and innovation program [45]. Beyond demonstration sites, there are initiatives such as EKLIPSE that aim to evaluate the performance and benefits of NBS [46].

*4.2. Multiplicity of Services and Supports*

4.2.1. A Major Advantage of NBS Compared to Traditional Solutions Is Their Multifunctionality

As mentioned by the definitions of IUCN [1] and the European Commission [2], NBS are multifunctional since they must simultaneously have benefits for humans and biodiversity. One interviewee puts it the following way: *"Often, the objective of NBS is to be multi-functional, and therefore should not meet only one environmental challenge. Otherwise, it is not called an NBS. At the very least, it must meet the objectives of adapting to climate change and bringing benefits to biodiversity. If, in addition, it can cool a place or store water, store carbon or be of recreational value to people, it is all for the better."* (A). This aspect is very important as the multifunctional aspect of NBS in cities can bring benefits to many fields such as (micro-)climate, ecology, hydrology, socioeconomics, land use planning, architecture, etc. In consequence, NBS can mitigate many risks such as flooding, heat waves or coastal erosion, etc. [47].

The hydrologist thinks that it is necessary to put forward the multifunctionality of the NBS for promoting them as follows: *"[there is not just one] service but several [ . . . ]. For example, a pipe handles water better than a green roof, but the roof has more benefits."* (C). He cited the following many functions delivered by NBS: *"Cooling, fighting against climate change, water management, human well-being, biodiversity conservation, city renaturation, and reconnecting with nature. For water management, an example of advantages is treatment at the source reduction of reject volumes in the networks, maximizing infiltration, minimizing runoff*

*in urban surfaces to avoid pollution transfer, limiting impacts on the natural environment."* (C). Other functions provided by NBS in the city, such as the case of wetlands, were mentioned as follows: *"living environment, depollution, shelter for aquatic fauna, maintenance of banks."* (D). The multifunctional aspect of NBS is one of its strongest points. For example, in a series of interviews conducted in Australia, 18 of 27 interviewees mentioned this aspect, and it was often compared to the single-benefit nature of grey infrastructures [27].

4.2.2. NBS Arch across Many Fields, therefore Transdisciplinarity and the Establishment of Network of Actors Must Be Encouraged

As suggested by their multifunctional aspect, the work on NBS in the city calls for many disciplinary fields. It is therefore essential to promote transdisciplinarity and to create networks between the different stakeholders. *"The importance of transversal governance is that it is very broad and that it associates the locals with the actors of the concerned territory in order to have a common co-constructed project"* (E). This collaborative governance is also one of the seven lessons put forward by Frantzeskaki [10] to improve the implementation of NBS in the city.

*"It is also necessary to make more co-constructions between the actors of the city and actors of biodiversity and of water. Because in fact, we often have projects done separately. We have our vision of things while the people working on the city have their own. Maybe creating a multidisciplinary working group on NBS would be interesting."* (A). The fact that professionals from different fields do not have the same ways of thinking refers to "silo thinking" and has already been identified as a barrier to green and blue infrastructure development [48].

It is important that actors working on urban NBS collaborate [19] and, in particular, make the link between operational needs and applied research, or the public and private sectors [22]. Public-private partnerships can even facilitate investments in NBS [36]. The need for transdisciplinary work, as well as the need to co-design NBS, have already been identified as a lever for NBS development [16,49]. Sarabi et al. [3] even identified stakeholder partnerships as an enabling factor for NBS in 27 papers. There are already collaborations between research and the operational field, such as one between École des Ponts ParisTech and the company SOPREMA, which works on insulation, waterproofing and roofs and offers a wide range of green roofs and facades.

To facilitate transdisciplinarity, it is important that professionals from various fields connect with each other [22]. On the brand "Végétal local" [50] (i.e., local vegetation), which sells wild local plants in France, an NBS representative stressed that it would be necessary *"to link the different initiatives like this one and to do everything together, while consulting each other and maintaining a good understanding between each other"* (B). The representative of the Regional Agency for Biodiversity explained her role as follows: *"To act as a link between all the actors, to help local authorities interested in carrying out NBS: they contact me and I guide them in finding technical support or consultancy firms that can carry out their field studies; I make sure that there is a follow-up of the projects, encourage them to find funding, and provide the means to carry out these projects. Interface role, lobbying for NBS."* (B) Such approaches should be developed in the future to better promote the NBS in the city. This interviewee also referred to the ARTISAN project already mentioned above and in which she is involved. Under ARTISAN and at the national scale, there is a *"resource network consisting of working groups of different themes for the production of resources and tools to support actors on NBS, develop a web interface, provide support for actors, support the mobilization of funding, initiate training programs, and develop studies for building indicators needed for the implementation of NBS, etc."* (B) At the regional level, the actors *"decide together to build a roadmap over 5 years period, and design a strategy for the development of NBS."* (B).

*4.3. Institutional and Financial Support*

4.3.1. What If NBS Could Save Money?

The ecologist from the Regional Biodiversity Agency had the opportunity to start a study to compare the costs of green infrastructures to those of grey infrastructures. This

study could not be completed due to a lack of data, but the preliminary results from a small sample showed that *"green infrastructures, in terms of investment and management, are less expensive than grey infrastructures. For example, underground concrete tanks for rainwater management are much more expensive than systems of swales, ponds or gardens."* (A). A more in-depth study is needed to confirm these results, but NBS are potential sources of financial savings. For example, a green roof can last twice as long as a traditional roof [51], thus saving money on roof renewal [52]. According to Fan et al. [53], the presence of NBS can even attract investments, while improving the image of the city that supports them. Green spaces that are part of NBS could also help attract knowledge professionals and participate in the city's economic development [54].

4.3.2. Have Special Funding for NBS

To compensate for the lack of financial means necessary for the development of NBS, one interviewee mentioned the following several ideas: *"Economic levers to facilitate investments in these NBS, at the community level are important. Perhaps, mechanisms like payments for ecosystem services, or favorable taxation on NBS are a plausible idea. Advantageous loans for communities willing to implement NBS are also a good idea."* (A). He also referred to the possibility of having funds from the European Commission for the development of NBS or at least for the restoration or the creation of ecosystems. In any case, it is very important that public and private funds become available for NBS in cities [55]. In the literature, there are different financial instruments that can encourage the use of NBS, such as the modification of user fees for ecosystem services, limiting impacts on natural areas and setting fiscal measures [36]. There may also be grants such as the one given by the European Social Fund to an NGO in Szeged, Hungary, for community gardens [56]. There are already initiatives to finance NBS, such as "Nature 2050" [57] (a program designed by CDC Biodiversity), which allows companies to fund NBS (some of which are in cities). The IUCN, in its 2018 French NBS committee brochure, indicates other ways to fund NBS, at least in France, such as funding for "climate" projects, "natural risk prevention" projects, calls for projects from water agencies, etc. [58].

4.3.3. Labels and Certifications That Endorse and Promote NBS

Labels and certifications can promote NBS in cities by enhancing their value. In this regard, an ecologist expressed his point of view about labels that, in his opinion, are closest to NBS, as follows: *"the BiodiverCity label on buildings. There are Ecojardin, EVE (Espace Végétal Écologique) labels for green spaces [ . . . ]. Labels are an opportunity for the development of NBS."* (A). The BiodiverCity label, supported by cibi (International Council on Biodiversity and Real Estate), concerns all urbanization projects in urban, peri-urban or natural sites, and displays the performance of real estate projects that take biodiversity into account [59]. Two of the goals referred to by NBS and set by this label are to "Maximize useful biotopes and ecological functionality" and to "Provide nature services for building users". One interviewee commented the following on this label: *"BiodiverCity provides a number of criteria to be fulfilled, among which the indigeneity of the plant palette, which demands to have a certain percentage of indigenous plants"* (I). A person working on the EcoQuartier label indicated that this label is obtained at the urban or rural project level. It includes the following four dimensions: approach and process, living environment and uses, territorial development, environment and climate. These dimensions contain 20 commitments. The following two of these commitments are linked to NBS: "Proposing urban planning to anticipate and adapt to climate change and risks" and "Preserve, restore and enhance biodiversity, soils and natural environments". These two commitments can therefore be used to implement NBS in the city. The presence of a green roof on a site can allow obtaining labels such as BiodiverCity, Effinature ("certification devoted entirely to the consideration of biodiversity in construction, renovation and development projects" [60]), or HQE (high environmental quality). However, it can also work the other way, as follows: Wanting one of these labels on a project can push them to implement a green roof. Currently, in French urban projects,

the label BiodiverCity seems to be the one that supports the most biodiversity and NBS in cities. However, according to several interviewees, this label does not demand enough from an ecological point of view. Experts [61] agree that other certifications also (BREEAM and LEED) incorporate NBS but not in sufficient detail, particularly with respect to vegetation. Therefore, it would be necessary to develop the existing labels and create new ones. A specific label for NBS could be beneficial.

4.3.4. Include NBS in the Regulations to Make Their Use Mandatory and Sustainable

Regulations are very important to promoting NBS [22]. According to the head of the town planning and natural environment department of the union for hydraulic development, it is even the best lever for setting up NBS in cities. She mentioned the Local Urban Plan (PLU; see Table 2) *"to have more nature in the city. This document allows for clearly defined orientations"* (D). This document has the power to impose NBS in cities. This union for hydraulic development also participates in the PLU by working with the municipalities to free up as much space as possible. An ecologist specified on this subject: *"with an urban planning document, one can make a zoning plan, let's say, for a wetland; one can impose the greening of roofs in the articles of the regulation; one can suggest the management of rainwater on the plot in green spaces; one can also suggest vegetated swales in front of buildings. In fact, you can do a lot of things with PLU."* (A). His colleague from the Regional Agency for Biodiversity explained as follows: *"We are trying to introduce often as we can in the PLU, PCAET (Territorial Climate-Air-Energy Plan) and in all other planning aspects the words NBS and measures for NBS."* (B). Water management at the parcel level can also be written into the PLU and hence promote NBS. Regarding NBS affecting water management, there are other French documents. One is called SDAGE (Water Development and Management Master Plan). Another one is named SAGE, which is a variation of SDAGE. This SAGE focuses more, at the local scale, on catchment areas and their watercourses. Since 2020, the hydraulic development union has been in charge of integrating the regulations of the SAGE with the *"implementation of protection measures, promoting non-imperviousness around the streets, and the infiltration of the first 8 mm of rain."* (D). Such regulations may encourage the use of NBS for urban water management. The registration of NBS in documents, such as the PLU ensures the durability of these solutions, beyond political changes (B). A hydrologist also cited the regulatory constraints by the communities that are in charge of sanitation such as the city of Paris (the departments, the water management union or the water agency), which can be *"either discharge limitation, or an abatement of so many mm of rainfall per event, rather than at the scale of a development or combination of several techniques, several NBS"* (C). Such documents can also promote the use of NBS in cities. In any case, it would be preferable for regulations to take into account the multi-benefit aspect of NBS [62].

**Table 2.** Names of French regulatory documents cited by interviewees (with their acronyms, full names in French and English, and definitions).

| Acronym | Full Name in French | Full Name in English | Definitions |
|---|---|---|---|
| PLU(i) | Plan Local d'Urbanisme (intercommunal) | Local (intermunicipal) Urban Plan | Main urban planning document at municipal or inter-municipal level. |
| PCAET | Plan Climat-Air-Energie Territorial | Territorial Climate-Air-Energy Plan | A planning tool that aims to mitigate climate change, develop renewable energy and control energy consumption. |
| SDAGE | Schéma Directeur d'Aménagement et de Gestion des Eaux | Water Development and Management Master Plan | Main tool for the implementation of the Community's water policy and set the guidelines for 6 years to achieve the objectives of "good water status". |
| SAGE | Schéma d'Aménagement de Gestion des Eaux | Water Management Plan | A more local version of the SDAGE, to reconcile the satisfaction and development of uses and the protection of aquatic environments. |

*4.4. Appropriation by the General Public*

4.4.1. Climatic Hazards Can Encourage the Use of NBS

Although climate can be considered a constraint for NBS, several interviewees said that they saw it more as an opportunity. Indeed, NBSs are often intended to manage the effects of climate change [23,63]. An ecologist declared the following: *"[the communities] will try to appropriate the subject. [ . . . ] if they see that there is flooding, they will want to deal with it"*, *"People are realizing that we should put back trees and vegetation, and that we should uncover rivers which have been buried in the recent past and so that they can play the role of sponges."* (A). This sponge notion refers to the "sponge city" concept that was brought forward by the Chinese government in 2013. It "describes an urban environment that is devoted to finding ecologically suitable alternatives to transform urban infrastructures into green infrastructures so these could capture, control and reuse precipitation in a useful, ecologically sound way." [64]. There are climate-related initiatives, such as the Life ARTISAN project already mentioned above, which deal only with the climate change adaptation facet of NBS. The fight against climate change may also represent an opportunity to include NBS in the regulations, as developed in the previous paragraph.

4.4.2. Raising Awareness and Communication to Re-Educate on Nature and Mainstream NBS

According to a water union official, *"It takes a lot of communication, for re-educating people about nature."* (D). *"There is also a need in governance, to have people in communities who are responsible for these NBS issues. That this concept must also be carried out at national level."* (A). There is a necessity to raise awareness in order to move away from the classical landscape approach that has subsisted for years and from the bad habit of using exotic plants (B). The representative of the Agency of Biodiversity proposed *"to go looking for the communities, to inform them, and to mainstream the term NBS"* (B). To mainstream the concept of NBS, media such as the internet, television, radio, and newspapers can be helpful [3]. It is also important to raise awareness about the effectiveness of NBS [45]. To have political and public support, awareness about the links between climate, health and the benefits of NBS is required [65]. The person responsible for the natural environment of the water union gave another piece of advice to educate the general public toward increased integration of nature into the city as follows: *"Create stages that would allow the eye to get used to the changes. For example, if you demolish a building, it attracts the eye, but if you remove the floors one by one, then you don't necessarily see that the building is being destroyed. With nature it's kind of the same, by going step by step, it can allow the uninitiated eye to better accept situations gradually rather than shocking it by a single event."* (D).

## 5. Conclusions

NBSs are considered an efficient tool to develop sustainable cities. For this reason, they are promoted by the UICN and the EU. By conducting several interviews with different professionals working on NBS (of academic, institutional and operational backgrounds), this study analyzed the barriers and levers related to their implementation in France. Despite belonging to different professional categories, the respondents seem to agree on common barriers affecting NBS implementation. The lack of scientific and cultural knowledge and the absence of financial, political, and institutional support appear to be the main reasons for the current low use of these solutions, whereas the technical problems raised by NBS seem to be more manageable.

To address these issues while putting more emphasis on the conservation of biodiversity, several avenues have been proposed by the interviewees. First, in order to improve the current knowledge of the functioning of NBS, some significant research efforts should be undertaken (essentially promoted by the academic and operational categories). They have to be carried out within a multidisciplinary framework. This seems necessary to better understand and assess the different ecosystem services (regulation, supply, cultural) provided by NBS, as they refer to the following several different disciplines: ecology, hydrology, mechanics, social sciences, urban planning, microclimate . . . It is also impor-

tant to have concrete demonstrative examples of NBS in cities that work well in order to highlight this multifunctional aspect (promoted by all categories). The follow-up and monitoring of these pilot sites will also contribute to producing quantitative data that will feed research activities.

The second track, which is also deeply related to the development of research, is education (which is promoted by all categories). The concept of NBS must be better and more widely introduced in training programs. The enrichment of knowledge will allow the consolidation of higher education courses and the promotion of their abilities to solve operational problems. The higher education framework will also facilitate the adoption of the multidisciplinary approach mentioned above and will promote the networking of the different actors involved in urban planning. The link between academic knowledge and its operational applications can also be performed by certifications and labeling that are commonly adopted by stakeholders. Obtaining a label or certification will eliminate any suspicion of "greenwashing" and also justify the implementation of an NBS rather than a traditional solution (essentially promoted by operational category). Note that this work has begun with the definition of the IUCN standard. These quantification tools could value NBS performances and also benefit from monitored pilot sites.

NBSs are also currently considered as climate change adaptation tools. In this application, they could be renamed NBAS (for Nature-Based Adaptation Solutions). In such an evolving climatic context, the sustainability of their ecosystem services and their associated performances over time is poorly known. This raises many questions about the choice of species to implement, their evolution in an urban environment strongly impacted by climate change, and their possible need for maintenance ... Here again, experimentation and the development of knowledge will provide answers that must be taken into account if we do not want to create new barriers to the implementation of NBS.

Finally, in addition to these scientific requirements, more social-societal initiatives can be proposed. Indeed, most of the seven lessons for planning NBS in cities proposed by Frantzeskaki [10] refer to consultation and co-construction approaches. The definition of new common goods, the involvement of the different local authorities, the appropriation of urban space by citizens, and the construction of a relationship of trust between these different entities, are all avenues to be favored to facilitate the implementation of NBS. This societal reorganization of urban planning, coupled with the democratization of knowledge, surely represents the best levers for overcoming the various obstacles listed in this article.

**Supplementary Materials:** The following supporting information can be downloaded at: https://www.mdpi.com/article/10.3390/su14169975/s1, Questionnaire used during the interviews.

**Author Contributions:** Conceptualization, C.D., N.F.-L. and P.-A.V.; methodology, C.D., N.F.-L. and P.-A.V.; formal analysis, C.D.; investigation, C.D.; writing—original draft preparation, C.D.; writing—review and editing, N.F.-L. and P.-A.V.; supervision, N.F.-L. and P.-A.V.; project administration, P.-A.V.; funding acquisition, P.-A.V. All authors have read and agreed to the published version of the manuscript.

**Funding:** This research was funded by Agence Nationale de la Recherche, grant number ANR-17-CE22-0002-01 EVNATURB project dealing with the evaluation of ecosystem performances for re-naturing urban environment. The APC was also funded by Agence Nationale de la Recherche, grant number ANR-17-CE22-0002-01 EVNATURB.

**Institutional Review Board Statement:** Ethical review and approval were waived for this study due to the interviews require only the information and consent of the interviewees. Informed consent was obtained verbally before participation.

**Informed Consent Statement:** Informed consent was obtained from all subjects involved in the study.

**Data Availability Statement:** Data sharing not applicable.

**Acknowledgments:** The authors would like to thank the different interviewees that accept to answer the questionnaire for their kind and constructive collaboration. We also want to thank Mario Al Sayah and Ghislaine Mangeney for correcting the English.

**Conflicts of Interest:** The authors declare no conflict of interest. The funders had no role in the design of the study; in the collection, analyses, or interpretation of data; in the writing of the manuscript, or in the decision to publish the results.

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
