# Peer review of "Barriers and Levers for the Implantation of Sustainable Nature-Based Solutions in Cities: Insights from France"

_sustainability, doi:10.3390/su14169975_

Round 1
Reviewer 1 Report
In this research, in an effort to identify the barriers and levers to the implementation of sustainable nature-based solutions in cities, several professionals working on NBS in France were interviewed; meanwhile, this study aims to identify these barriers are and to highlight the levers that can be used to overcome them. Basically, this paper may meet the subject and level of requirements of a "sustainable" journal. However, I sincerely recommend that the authors refer to the following review comments to revise the paper.
1. This paper mentions "In urban environment, the most cited NBS are: urban forests, green roofs and walls, ecological corridors, or green swales. They are usually studied to manage stormwater management issues or mitigate urban heat islands.", basically, the author should provide relevant data or literature as supporting evidence.
2. Since the twelve respondents came from three professional categories (Operational/Academic/Institutional), are there differences in their professional analysis perspectives?
3. It is mentioned in the article that "NBS are related to projects, roofs, communities, regulations, services, people, etc.", the author can make more statements and explanations about this important discovery.
4. The study interviewed 12 professionals and summarized several key findings. However, a comprehensive discussion of these research findings with relevant literature is recommended to strengthen the credibility of the conclusions obtained.
5. Authors should provide compiled manuscripts to reduce dyslexia during review. In fact, many deleted and modified contents appeared repeatedly, causing great trouble for reviewer to read.
Author Response
1-This paper mentions "In urban environment, the most cited NBS are: urban forests, green roofs and walls, ecological corridors, or green swales. They are usually studied to manage stormwater management issues or mitigate urban heat islands.", basically, the author should provide relevant data or literature as supporting evidence.
Some references have been added to justify this list of NBS found in urban environment.
2-Since the twelve respondents came from three professional categories (Operational/Academic/Institutional), are there differences in their professional analysis perspectives?
Indeed, the 12 interviewees come from different professional categories. Despite this difference, the respondents seem agree to common barriers affecting NBS implementation. The difference was more visible concerning the levers. Research efforts promoted by the academic and operational categories, the development of demonstrative sites promoted by all categories. These details have been added in the Conclusion section.
3- It is mentioned in the article that “NBS are related to projects, roofs, communities, regulations, services, people, etc.”, the author can make more statements and explanations about this important discovery.
The lists from word clouds have been completed at the beginning of Sections 3 and 4 with some references to the sub-sections where explanation are provided and illustrated.
4- The study interviewed 12 professionals and summarized several key findings. However, a comprehensive discussion of these research findings with relevant literature is recommended to strengthen the credibility of the conclusions obtained.
The results and analysis made from the interviews are presented and discussed regarding existing relevant literature. For example:
- the multiplicity of terms designing NBS: ref [16]
- the lack of knowledge related to NBS: ref [17,18]
- the necessity to develop assessment indicators: ref [18]
- the need to have technical guidelines: ref [20]
- the problem of NBS maintenance: ref [20]
- the challenge to adapt the vegetation to climate change: ref [16,23]
- the problem of space in urban environment: ref [26]
- the perception of NBS as greenwashing: ref [27]
- the absence of biodiversity in our societal debates: ref [29]
...
5- Authors should provide compiled manuscripts to reduce dyslexia during review. In fact, many deleted and modified contents appeared repeatedly, causing great trouble for reviewer to read.
The corrected manuscript has been completely revised to avoid such repetitions and typos.
Reviewer 2 Report
The authors have chosen the Nature-Based Solutions in cities as the main theme of their paper.
It was well mentioned that this particular type of research is not always accepted as credible and often is accepted as a "buzz-word" only. In the Metholody section, the reviewer has seen that the authors tried to correct the poll of responders - whereas it is understood that the aim was to have a variety of responders, there is no single key as to why and how they were chosen.
Additionally, due to their different positions and background, the outcomes are not consistent since the responses can vary based on the education, age structure or professional position. Only 12 people were interviewed - hence the stastics probability of the outcomes might not be true, and since the questions were not presented - therefore part of the methodology is missing.
The authors quote some of the responses - but are these wholesome, since they are stated by a single person only?
Author Response
The authors have chosen the Nature-Based Solutions in cities as the main theme of their paper.
It was well mentioned that this particular type of research is not always accepted as credible and often is accepted as a "buzz-word" only. In the Methodology section, the reviewer has seen that the authors tried to correct the poll of responders - whereas it is understood that the aim was to have a variety of responders, there is no single key as to why and how they were chosen.
Indeed, the aim was to have a variety of responders. Interviewees were chosen to have equal representation from the academic, institutional and operational worlds. The respondents were selected because the authors knew that these people or their organizations/companies were working on urban NBS in France for several years, and they were able to share some relevant and eclectic experiences, discourses and analyses. The authors are aware that this selection may have led to a bias, but as this work is rather qualitative, this bias may be negligible. This methodological element has been added in Section 2.
Additionally, due to their different positions and background, the outcomes are not consistent since the responses can vary based on the education, age structure or professional position.
The aim of the study is to identify a maximum of barriers and levers to Nature-based Solutions in the city, thanks to the people who work directly on this topic in France. This is why we interviewed people with very different positions and backgrounds. We did not seek to have homogeneous answers, but to have various subjective points of view on the subject. The answers may indeed depend on education, age or profession, but they are all based on a quite significant work experience on this topic. This has been recalled in the Methodology section.
Only 12 people were interviewed - hence the stastics probability of the outcomes might not be true,
Indeed, only 12 people were interviewed, which does not allow for real statistical analysis, but the study is qualitative and not quantitative. This is why the interview was semi-directive with open-ended questions. If we had wanted to do a quantitative study, we would have interviewed more people in a shorter format with more closed questions. The size of the sample could have been higher in the case of a random selection for example. Moreover, the interviewees were chosen because of their knowledge on the NBS topic, as this study was focused on the barriers and levers identified by experimented persons. It did not aim to assess the level of knowledge of general public.
and since the questions were not presented - therefore part of the methodology is missing.
The interview questions have been placed in the Supplementary Materials, but the main questions about barriers and levers to NBS have been added to the methodology section.
The authors quote some of the responses - but are these wholesome, since they are stated by a single person only?
Quotations from interviewees' responses is a way of presenting social science findings that keeps them true to what they say. For example, see: Corden, A., & Sainsbury, R. (2006). Exploring ‘quality’: Research participants’ perspectives on verbatim quotations. International Journal of Social Research Methodology, 9(2), 97-110. These quotations refer to some personal points of view. Contextualized and analyzed using bibliographic references or additional discourses, they shed light on a given subject. This explanation has been added in the Methodology section.
Round 2
Reviewer 1 Report
The author has adjusted the article according to the review comments, and the content of the article has met the level of the journal.
It is suggested that the abstract of the article can be adjusted appropriately, and information of important research results can be added to enhance the reference value of reading.
Reviewer 2 Report
The authors have responded to all of the questions. The paper still includes a certain level of ambiguity, but this might be due to the small sample of respondents used by the authors. The review agrees that the paper can be published in its present form.
Author Response
Please see the attachment.

This manuscript is a resubmission of an earlier submission. The following is a list of the peer review reports and author responses from that submission.
Round 1
Reviewer 1 Report
Authors give NBS solutions but it would be helpfull for the reader to give a better overview of the literature regarding the topic, not just number of publications. After reading Introduction I have no clear image what NBS is. Few examples would be a welcome addition.
Regarding Metodology, I find the survey poorly constructed. Also, term "actor" is not apropriate term for their role. Table 1. needs editing.
Figure 1. Authors provide word clouds with answers but give only vague mention about the questions itself. It is actually very confusing.
The chapters are inconsistent mixture of quotes and text written by authors, too may subchapters makes the text and research very difficult to follow. Overall, this is more an article for newspapers, than for a scientific journal.
Reviewer 2 Report
Nature-based solutions are proliferating in European cities over the past years as viable solutions to urban challenges such as climate change and the loss of biodiversity. Nature-based solutions require collaborative governance.
Moreover an inclusive narrative of mission for nature-based solutions can enable integration to many urban agendas.
The presented manuscript has a well-defined topic, which has been analyzed consistently. The methodology was quite well prepared. The authors rightly divided the chapters into points, which makes the work readable.
Possibly, I would add a practical item to the discussion:
Seven lessons for planning nature-based solutions in cities https://doi.org/10.1016/j.envsci.2018.12.033
However, in my opinion, the conclusions are too general, easy to predict, not innovative. The Discussion section is minimalistic and showing just generally known information.